# The Oral–Brain Axis in Alzheimer’s Disease: From Microbial Dysbiosis to Neurodegeneration

**DOI:** 10.3390/microorganisms13122741

**Published:** 2025-12-01

**Authors:** Alessia Felicetti, Domenico Azzolino, Pietro Paolo Piro, Gabriel César Dias Lopes, Najmeh Rezaeinezhad, Roberto Lovero, Luisella Bocchio-Chiavetto, Marica Colella, Pier Carmine Passarelli

**Affiliations:** 1Department of Medical and Surgical Sciences, Magna Graecia University, 88100 Catanzaro, Italy; felicettialessia21@gmail.com; 2Geriatric Unit, Medical Department, Fondazione IRCCS Ca’ Granda Ospedale Maggiore Policlinico di Milano, 20122 Milan, Italy; do.azzolino@hotmail.it; 3Fondazione Policlinico Universitario A. Gemelli IRCCS, Catholic University of Sacred Hearth, 00168 Rome, Italy; pierpaolopiro1@yahoo.it; 4Department of Neuroscience and Mental Health, School of Science of Health, Logos University International (UNILOGOS), Miami, FL 33137, USA; president@unilogos.edu.eu; 5Department of Neuroscience and Mental Health, School of Science of Health, European International University, 75018 Paris, France; 6Department of Exercise Physiology, Faculty of Sport Sciences and Health, University of Tehran, Tehran 1439957131, Iran; 7Clinical Pathology Unit, AOU Policlinico Consorziale di Bari-Ospedale Giovanni XXIII, 70124 Bari, Italy; robertolovero69@gmail.com; 8Department of Theoretical and Applied Sciences (DiSTA), eCampus University, 22060 Novedrate, Italy; luisella.bocchiochiavetto@uniecampus.it; 9IRCCS Centro San Giovanni di Dio Fatebenefratelli, 25125 Brescia, Italy; 10Section of Microbiology and Virology, Interdisciplinary Department of Medicine, School of Medicine, University of Bari, 70124 Bari, Italy; 11Department of Head and Neck and Sensory Organs, Division of Oral Surgery and Implantology, Fondazione Policlinico Universitario A. Gemelli IRCCS, Catholic University of Sacred Hearth, 00168 Rome, Italy; piercarminepassarelli@hotmail.it

**Keywords:** oral dysbiosis, systemic health, Alzheimer’s disease

## Abstract

Alzheimer’s disease (AD), the most prevalent form of dementia, still lacks a clearly defined pathogenesis and effective disease-modifying therapies, prompting growing interest in peripheral drivers of neurodegeneration. Among these, chronic oral dysbiosis has emerged as a potential risk factor. Disruption of the oral ecosystem in periodontitis promotes systemic inflammation and the circulation of bacterial products capable of influencing brain homeostasis. By integrating molecular findings with epidemiological data linking periodontitis, tooth loss, and poor oral health to increased AD risk, this review examines how oral dysbiosis contributes to systemic inflammation as part of a broader network of interacting factors involved in AD pathophysiology. It describes how inflammatory, gut-microbial, genetic, and barrier-related processes intersect with oral dysbiosis and jointly contribute to the acceleration of AD progression. Building on this systemic perspective, the review highlights emerging oral biomarkers and oral–gut microbiota-targeted therapies as potential tools to address current gaps in early diagnosis and intervention. Overall, this work advances current understanding by integrating previously fragmented evidence and highlighting the key conceptual and methodological gaps that must be addressed to clarify causality and to guide the development of preventive and therapeutic approaches targeting oral health in the context of AD.

## 1. Introduction

Alzheimer’s disease (AD) is the most common neurodegenerative dementia, characterized by progressive memory loss, cognitive decline, and behavioral changes, with profound effects on patients and their families [1]. Despite decades of research, currently available therapies remain limited and fail to significantly modify disease progression, while its etiopathogenesis continues to be debated [2]. Traditionally, research has centered on the amyloid hypothesis, which attributes neurodegeneration to the accumulation of β-amyloid protein (Aβ), extracellular plaque formation, and tau hyperphosphorylation leading to neurofibrillary tangles [3]. However, emerging evidence suggests that these processes may be triggered or amplified by external factors, particularly chronic peripheral inflammation. Such systemic inflammatory processes are increasingly understood as part of a broader network linking host immunity, microbiome alterations, and brain vulnerability. Within this framework, the microbiome–gut–brain axis has emerged as a key conceptual model, illustrating how microbial communities throughout the body interact with immune, metabolic, and neural pathways to influence neurodegenerative processes [4]. Alterations in peripheral microbial ecosystems, whether intestinal or oral, can enhance systemic cytokine levels, modify barrier integrity, and affect central nervous system (CNS) homeostasis. Consequently, oral dysbiosis represents one component of a wider microbiome-driven inflammatory network with potential relevance to AD [5]. Increasing attention has therefore been directed toward oral health, specifically microbiota dysbiosis and periodontitis, as modifiable contributors to AD risk [6,7,8]. The oral cavity harbors more than 700 microbial species in a dynamic equilibrium. In conditions such as periodontitis, this balance is disrupted, leading to chronic infection, bacterial translocation, and the systemic spread of endotoxins [9,10,11]. Such conditions have also been proven to contribute to oral cancer through the pro-inflammatory activity of certain members of the oral microbiota, in particular *Porphyromonas gingivalis*, *Fusobacterium nucleatum*, and *Candida albicans* [12]. *Porphyromonas gingivalis*, a Gram-negative rod-shaped bacterium, has emerged as a “keystone pathogen”, capable of altering immune responses and invading brain tissue, where its lipopolysaccharides (Pg-LPSs) and gingipain proteases activate microglia and astrocytes, promoting neuroinflammation, oxidative stress, and synaptic loss, as well as alter the basic neurological functions, i.e., body temperature control [13,14,15]. Similarly, *Fusobacterium nucleatum* has been shown to exacerbate Aβ accumulation and tau phosphorylation in animal models [16]. Epidemiological studies reinforce this association: large cohorts, such as that analyzed by Kulkarni et al. [17] using the TriNetX database, reveal that poor oral health nearly doubles AD risk, with tooth loss identified as one of the most consistent predictors [18,19]. Experimental data confirm these findings, showing that tooth extraction or dental pulp loss in animal models accelerates neuronal degeneration along the trigeminal–locus coeruleus–hippocampus axis, directly impairing memory [20]. Recent research further highlights the role of the oral microbiota within the broader oral–gut–brain axis, where bacterial metabolites, extracellular vesicles, and inflammatory mediators increase blood–brain barrier (BBB) permeability and trigger glial activation [4,5,21]. Within this framework, the oral microbiome emerges not only as a pathogenic factor but also as a potential early biomarker [22,23]. Saliva has gained attention as a diagnostic fluid: salivary biomarkers such as lactotransferrin, multi-omics signatures, and oral microbial profiles have been associated with AD risk and progression [24,25,26]. Although standardization is still lacking, these approaches pave the way for non-invasive early diagnostic strategies. Collectively, epidemiological, experimental, and molecular evidence supports the existence of a true mouth–brain axis, in which oral dysbiosis and periodontitis function not merely as comorbidities, but as active pathogenic drivers influencing the onset and progression of Alzheimer’s disease [27,28].

## 2. Materials and Methods

This narrative review synthesizes current evidence on the role of the oral microbiota and periodontal disease as modifiable risk factors for Alzheimer’s disease, with emphasis on mechanistic pathways, biomarkers, and therapeutic approaches. A literature search was performed in PubMed, Scopus, and EMBASE up to October 2025 using controlled vocabulary and free-text terms (oral microbiome, periodontal disease, dysbiosis, *Porphyromonas gingivalis*, neuroinflammation, Alzheimer’s disease, biomarkers, therapeutics). Eligible studies included original research and reviews in English addressing oral microbiota or periodontal disease in relation to AD or cognitive decline, with a focus on mechanisms, biomarkers, or interventions. Both preclinical and clinical studies were included, while case reports, abstracts, and non-peer-reviewed sources were ruled out. The final selection of studies was organized thematically to provide a comprehensive overview of the state of the science and to highlight gaps warranting future investigation. As this is not a systematic review, no formal inclusion/exclusion protocol or PRISMA flowchart was used.

## 3. Oral Microbiome: From Symbiosis to Dysbiosis

### 3.1. The Oral Microbiome

The oral cavity harbors one of the most complex microbial ecosystems in the human body, comprising more than 700 bacterial species that coexist in a dynamic equilibrium with the host [29]. In a healthy state, this community is largely characterized by symbiotic or commensal relationships, where resident microbes contribute to oral homeostasis and protect against the overgrowth of pathogens [30,31]. The core microbiota includes Gram-positive cocci such as *Streptococcus mitis*, *Streptococcus sanguinis*, *Streptococcus oralis* and *Rothia dentocariosa*, Gram-positive bacilli including *Actinomyces* spp. and Gram-negative cocci such as *Veillonella parvula* and *Neisseria* spp. [32]. These organisms perform essential ecological functions, ranging from competition for adhesion sites to the modulation of host immunity, thereby constituting the first line of defense against exogenous microbes [33,34].

However, the oral microbiome is not static. Environmental factors such as age, pregnancy, smoking, alcohol consumption, diet, saliva composition, systemic health, and hygiene practices influence its composition and functional balance [34]. When stability is maintained, commensal species, like *Streptococcus gordonii*, *S. parasanguinis*, and *S. oralis,* can even antagonize pathogens such as *Streptococcus mutans*, reducing the risk of caries [35,36]. Likewise, *Streptococcus cristatus* suppresses virulence gene expression in *Porphyromonas gingivalis*, limiting its pathogenic potential [37]. Nevertheless, under conditions of host compromise or ecological disturbance, commensals may acquire opportunistic behavior. For instance, cooperative interactions between *S. gordonii* and *P. gingivalis* exacerbate alveolar bone loss, while streptococcal metabolites can modulate *P. gingivalis* attachment and virulence [38,39]. Similarly, *S. gordonii* and *S. parasanguinis* can enhance the pathogenicity of *Aggregatibacter actinomycetemcomitans*, and streptococcal species such as *S. oralis* and *S. sanguinis* may potentiate the virulence of *Candida albicans* [40,41].

These complex multispecies interactions highlight the dual role of the oral microbiome: protective in a state of balance, but capable of shifting toward dysbiosis under ecological or host-related perturbations. In this context, commensals can act as “accessory pathogens”, cooperating with keystone organisms such as *P. gingivalis* or *A. actinomycetemcomitans* to drive disease processes, particularly periodontitis [42,43,44]. Recognizing this dynamic continuum, from symbiosis to dysbiosis, is essential for understanding how the oral microbiota shapes both local periodontal outcomes and systemic health [45,46].

### 3.2. The Periodontal Microbial Complex Theory

Microbial dysbiosis in periodontitis refers to a disruption of the oral microbiome’s balance, marked by altered species proportions and functional activity. This ecological shift moves the community from a health-associated state dominated by Gram-positive commensals to a disease-associated state enriched in Gram-negative anaerobes. The resulting increase in virulence factor expression fuels host inflammation and periodontal tissue destruction [47,48]. To understand the microbial ecology of periodontal diseases through the application of cluster analysis, Socransky et al. [49] introduced a paradigm-shifting framework which enabled the classification of subgingival species into distinct microbial complexes according to their patterns of co-occurrence and associations with clinical parameters of disease (Figure 1). Among these, the red complex, comprising *Porphyromonas gingivalis*, *Tannerella forsythia*, and *Treponema denticola*, demonstrated the strongest correlation with advanced periodontal pockets and clinical attachment loss, thereby establishing its central role in disease severity. The orange complex, which included species such as *Fusobacterium nucleatum*, *Prevotella intermedia*, and *Campylobacter rectus*, was characterized as an ecological bridge, facilitating the transition from health-associated microbiota to pathogenic consortia. By contrast, the yellow, green, and purple complexes, largely consisting of early colonizers such as *Streptococcus*, *Capnocytophaga*, *Actinomyces*, and *Veillonella* species, were associated with periodontal health and the initial stages of biofilm development (Figure 1).

This model was critical in demonstrating that periodontal pathogenesis is not attributable to a single organism but instead arises from synergistic interactions among microbial consortia, in which microbial-complex shifts promote pathogenicity [50]. By altering the subgingival microenvironment through metabolic activity, inflammatory stimulation, and co-aggregation properties, orange complex species such as *Prevotella intermedia*, *Campylobacter rectus*, and *Fusobacterium nucleatum* create conditions favorable for the colonization and persistence of highly pathogenic red complex bacteria, including *Porphyromonas gingivalis*, *Tannerella forsythia*, and *Treponema denticola* [51]. Such microbial interactions set the stage for oral dysbiosis: a state that not only drives local periodontal destruction but also promotes systemic immune activation through the release of cytokines, LPSs, outer membrane vesicles (OMVs), proteolytic enzymes, and metabolites with immunomodulatory properties. Circulating microbial components and inflammatory mediators can modulate gut microbial communities, promote intestinal dysbiosis, and disrupt epithelial tight junctions, thereby increasing systemic permeability and facilitating translocation of virulence factors [4,52,53]. This cascade contributes to endothelial dysfunction and increased BBB permeability, enabling microbial-derived molecules, including Pg-LPSs, gingipains, and OMVs, to access the central nervous system and trigger microglial priming, neuroinflammation, and oxidative stress [4,5,21,54,55,56,57,58]. Metabolomic alterations associated with oral dysbiosis, including shifts in short-chain fatty acids and amino acid-derived metabolites, further influence host immune tone and may amplify neuroinflammatory signaling along the oral–gut–brain axis [22,59]. Collectively, these processes highlight the bidirectional nature of microbiome–brain signaling, whereby oral microbial imbalance perturbs systemic homeostasis and contributes to neurodegenerative vulnerability. This integrated perspective places the periodontal microbiota within a broader multisystem network linking microbial ecology, immune activation, barrier dysfunction, and CNS pathology [4,5,21,22,23].

### 3.3. Oral Dysbiosis at the Crossroads of Inflammation, Immunity, and Neurodegeneration

The pathogenic potential of the oral microbiome extends far beyond the confines of the periodontium. Once considered primarily as agents of local tissue destruction, organisms of these complexes are now recognized as systemic modulators with far-reaching implications for human health. Among them, *Aggregatibacter actinomycetemcomitans* promotes colonization by cohabiting bacteria and undermines host immune defenses through the activity of cytolethal distending toxin, thereby sustaining chronic inflammation [60]. *Prevotella intermedia*, traditionally associated with periodontitis, has also been linked to extra-oral diseases such as cystic fibrosis, chronic bronchitis, and atherosclerosis. In vitro evidence demonstrates that *P. intermedia* lipopolysaccharides (Pg-LPSs) induce nitric oxide and interleukin-1β (IL-1β) production in murine macrophages, thereby driving pro-inflammatory cascades [61]. Furthermore, its virulence arsenal, comprising LPSs and cysteine proteases, enhances immune evasion and sustains chronic inflammation [62].

The systemic implications of oral pathogens extend across multiple organ systems. Periodontal infections have been associated with cardiovascular disease, with population-based studies showing a significant correlation between poor oral health and vascular pathology, underscoring oral microbiota as an independent risk factor for heart disease [63]. Comparable mechanisms underlie adverse pregnancy outcomes, as translocated oral pathogens and their inflammatory products compromise maternal–fetal immune balance [64]. In autoimmune contexts, *P. gingivalis* has been implicated in the loss of immune tolerance through protein citrullination, providing a plausible mechanistic link between periodontitis and rheumatoid arthritis [65].

Emerging evidence indicates that dysbiosis in the oral microbiota, such as seen in periodontitis, can alter gut microbiome composition, degrade gut barrier integrity, and increase systemic exposure to bacterial products (e.g., LPS) and inflammatory mediators, thereby contributing to chronic inflammation and metabolic disturbances. For instance, studies have found that oral bacteria commonly swallowed in saliva can ectopically colonize the gut, leading to elevated circulating cytokines, impaired mucosal immunity, and shifts in microbial metabolites that affect insulin resistance and cardiovascular risk [21,22]. Clinical investigations also associate periodontal disease with gut microbiota changes, endotoxemia, and systemic markers of inflammation in conditions like inflammatory bowel disease, obesity, and nonalcoholic fatty liver disease, underscoring the oral–gut axis as a potential driver of systemic disease burden [52,53].

Research has also linked the oral microbiome to cancer pathogenesis and metabolic disorders. Dysbiosis marked by enrichment of genera such as *Rothia*, *Haemophilus*, *Corynebacterium*, *Paludibacter*, *Porphyromonas*, *Oribacterium*, and *Capnocytophaga* distinguishes patients with oral and oropharyngeal cancers from healthy controls with high diagnostic accuracy, supporting the utility of microbial signatures as non-invasive biomarkers [66]. Oral pathogens have also been implicated in gastrointestinal malignancies: *Porphyromonas* species contribute to esophageal cancer [67], while alterations in the oral microbiome are now emerging as potential biomarkers and therapeutic targets in pancreatic cancer, possibly through chronic systemic inflammation and modulation of host immunity [68]. Moreover, the capacity of *P. gingivalis* OMVs to deliver gingipains to the liver and impair insulin signaling highlights a mechanistic link between oral bacteria and metabolic disorders such as diabetes [69].

In recent years, oral dysbiosis has emerged as a potential contributor to neurodegenerative disorders, most notably Parkinson’s disease [70,71] and Alzheimer’s disease [72,73]. A growing body of evidence suggests that periodontal pathogens may influence brain health through systemic inflammation, disruption of the Blood–Brain Barrier, and the release of neurotoxic virulence factors such as gingipains. These insights point to the oral microbiome as a possible mediator between oral health and CNS pathology, offering a new perspective on the mechanisms underlying dementia.

## 4. Oral Dysbiosis in Alzheimer’s Disease: Linking Microbial Imbalance to Neurodegeneration

### 4.1. Oral Microbiota at the Interface of Neuroinflammation and Alzheimer’s Disease

Chronic neuroinflammation is increasingly recognized as a central mediator linking oral dysbiosis to AD pathology. Periodontal pathogens, including *P. gingivalis* and *F. nucleatum*, promote systemic elevation of IL-1β, IL-6, and Tumor Necrosis Factor-α (TNF-α) [74,75]. These cytokines are capable of crossing or disrupting the blood–brain barrier, where they trigger microglial priming and sustained activation of innate immune pathways. As demonstrated by Lue et al. [76], activated microglia release reactive oxygen and nitrogen species, complement proteins, and additional pro-inflammatory mediators that amplify amyloid-β deposition and tau phosphorylation, accelerating synaptic failure and neuronal death.

Virulence factors such as LPS, OMVs, and gingipains released by *P. gingivalis* provoke systemic immune activation and inflammatory responses that exacerbate neuroinflammation, contributing to the progression of Alzheimer’s disease and related dementias [54,55,60]. *P. gingivalis* lipopolysaccharides, a principal component of the bacterial outer membrane, are delivered to neuronal cells via OMVs through direct membrane fusion or endocytosis [56,57]. Pg-LPSs are then internalized via receptor-mediated endocytosis, engaging pattern recognition receptors such as Toll-like receptor 4 [58]. Human caspase-4 and murine caspase-11, homologous proteins, are activated upon cytosolic detection of internalized LPSs, initiating the noncanonical inflammasome pathway [77,78,79]. Under conditions of low Aβ concentrations, typical of the early stages of AD, binding of caspase-4 to Pg-LPSs induces its activation and polymerization, subsequently cleaving gasdermin D (GSDMD) to produce the N-terminal fragment (GSDMD-N). GSDMD-N inserts into the plasma membrane, forming pores that induce pyroptosis, a form of programmed cell death, and K^+^ efflux [80]. The resulting ionic flux activates the NLRP3 inflammasome, culminating in caspase-1 activation, IL-1β maturation, and secretion [81,82]. Caspase-4-dependent inflammasome activation also enhances reactive oxygen species (ROS) generation, further promoting NLRP3 hyperactivation and inflammatory signaling [83,84]. Concurrently, Pg-LPSs stimulate the GSK3β pathway, a mechanism already implicated in tau phosphorylation, thereby providing a direct link between periodontal infection, synaptic loss, and cognitive impairment [14]. *P. gingivalis* and its gingipain enzymes can proteolytically modify plasma lipoproteins, including apoE and apoB-100 [85]. Given the central role of apoE in amyloid-β metabolism and clearance [86], such modifications could impair lipid transport in the brain and promote amyloid accumulation, linking periodontal infection to AD pathology. This framework has been reinforced by evidence that bacterial gingipain proteases promote both Aβ accumulation and tau hyperphosphorylation, amplifying the classical neuropathological cascade of Alzheimer’s disease [3,15]. Furthermore, gingipain activity induces structural and functional alterations in lipoproteins that may exacerbate neuroinflammatory processes. These events are accompanied by increased ROS production, contributing to oxidative stress, neuronal damage, and the progression of AD-related neurodegeneration [85].

In parallel, *F. nucleatum*, a Gram-negative anaerobic fusiform bacterium frequently associated with periodontitis, has been shown to exacerbate cognitive decline and increase Aβ and phosphorylated tau deposition in transgenic mouse models [16,87]. Taken together, these findings indicate that multiple oral pathogens may function as triggers or accelerators of neurodegeneration, linking periodontal disease with the progression of Alzheimer’s pathology [50].

### 4.2. Oxidative Stress and Cell Death

Neurodegeneration and AD pathology involve multiple cellular disturbances, notably through mitochondrial impairment that disrupts ATP production, elevates oxidative stress, and promotes excessive ROS generation, ultimately contributing to neuroinflammation and neuronal death [88,89,90]. Beyond their canonical role in bioenergetics, mitochondria are critical regulators of various non-energetic cellular processes, including the intrinsic apoptotic pathway [91]. During *P. gingivalis* infection, mitochondria have been implicated in modulating apoptosis across multiple host cell types, which may facilitate tissue damage and bacterial immune evasion [92]. Moreover, mitochondria contribute to intercellular signaling by releasing ROS and mitochondrial DNA (mtDNA), both acting as potent signaling molecules during infection [92]. Notably, *P. gingivalis* has been shown to influence mitochondrial ROS production, thereby modulating downstream inflammatory signaling pathways [93].

Numerous studies have also demonstrated that periodontitis is associated with elevated markers of oxidative damage, such as malondialdehyde (MDA) and 8-hydroxy-2′-deoxyguanosine (8-OHdG), together with a reduction in systemic antioxidant defenses [94,95,96,97,98,99,100,101,102]. These studies consistently show that patients with periodontitis exhibit elevated markers of oxidative damage, including increased salivary, serum, and gingival crevicular fluid levels of 8-OHdG (a marker of DNA oxidation) [96,97,98,99,102] and MDA (a marker of lipid peroxidation) [95,100,101]. Such biomarkers not only correlate with disease severity but also decline following periodontal therapy, highlighting oxidative stress as both a hallmark and a modifiable component of periodontal pathogenesis. Beyond classical oxidative stress, the recent literature has introduced ferroptosis as a novel connection between periodontal disease and systemic conditions, including Alzheimer’s disease [103]. Chronic inflammation triggers iron accumulation and lipid peroxidation, promoting ferroptosis, a form of iron-dependent cell death that in periodontal tissues affects fibroblasts and osteocytes. Experimental evidence shows that human gingival fibroblasts exposed to Pg-LPSs show mitochondrial alterations, iron overload, lipid peroxidation, and elevated levels of MDA, and a reduction in antioxidant defenses such as GPX4 [104]. Consistently, in vivo models of periodontitis display the same ferroptotic markers, while pharmacological inhibition of ferroptosis alleviates tissue damage, underscoring its pathogenic role in periodontal destruction [104].

### 4.3. Blood–Brain Barrier Dysfunction

Another critical link between the mouth and brain is represented by the blood–brain barrier (BBB). Numerous in vitro experiments have shown that infection of cerebral endothelial cells with *P. gingivalis* induces cytotoxicity, cell-cycle arrest, and the release of pro-inflammatory cytokines [105,106]. This process compromises BBB integrity and facilitates the entry of pathogens and inflammatory mediators into the brain parenchyma. More recent studies have highlighted the role of OMVs, nanosized extracellular vesicles either secreted by oral bacteria or released by infected host cells, in mediating this process both in vitro and in vivo [107]. These vesicles, loaded with bacterial antigens and inflammatory mediators, preferentially accumulate in hippocampal microglia, thereby promoting neuroinflammation.

Notably, *P. gingivalis* OMVs enriched in gingipain proteases exert multiple neurotoxic effects. They can degrade tight junction proteins such as zonula occludens-1 (ZO-1), occludin, and claudin-5 in human cerebral microvascular endothelial cells (hCMEC/D3), thereby exacerbating BBB permeability [108,109]. In murine models, repeated systemic or oral exposure to *P. gingivalis* OMVs leads to their rapid translocation into cortical and hippocampal regions [110], accompanied by astrocyte and microglial activation, IL-1β release, tau hyperphosphorylation, and cognitive decline [111]. Moreover, gingival exposure to *P. gingivalis* or its OMVs has been shown in murine models to induce periodontitis, systemic inflammation, and neuroinflammatory responses, ultimately resulting in memory impairment-like behaviors [112].

### 4.4. Amyloidogenesis and Cross-Seeding

The accumulation of amyloid-β (Aβ) plaques and tau tangles represents a defining feature of AD, and neuroinflammation induced by microbial components such as Pg-LPSs has been shown to accelerate these pathological processes [113,114,115,116]. Experimental studies demonstrate that *P. gingivalis* or its LPSs promote amyloidogenic processing of the amyloid precursor protein (APP), enhancing Aβ generation through the β- and γ-secretase pathways [106,117,118,119]. Within this cascade, presenilin-1 (PS1), a γ-secretase subunit, plays a pivotal role in producing the neurotoxic Aβ1–42 isoform [120]. In APP/PS1 models, the extracellular matrix protein reelin has been observed to co-localize with Aβ plaques [121,122] and to act as a molecular bridge between Aβ and tau, thereby linking amyloid and tau pathologies [123]. The interplay among APP, PS1, reelin, Aβ, and tau may therefore constitute a critical axis driving neurodegeneration and AD progression. A further line of research suggests that oral biofilms themselves may serve as peripheral reservoirs of amyloid-β. Ex vivo analyses of biological samples have detected Aβ in dental tissues and periodontal biofilms, showing that amyloid production is not exclusively a cerebral phenomenon [124]. This peripheral accumulation may contribute to a “cross-seeding” process with cerebral amyloid, enhancing pathological deposition. In addition, proteins from both oral and non-oral pathogens have been shown to directly modulate Aβ aggregation, promoting toxic fibril formation or APP accumulation [125]. In APP/PS1 transgenic mice, experimental periodontitis was shown to exacerbate Alzheimer-like pathology by aggravating cognitive deficits, increasing Aβ deposition, enhancing microglial activation, and disrupting both gut microbial balance and intestinal/brain barrier integrity [126]. Similarly, transplantation of salivary microbiota from periodontitis patients into APP/PS1 mice aggravated amyloid accumulation, neuroinflammation, and cognitive impairment, concomitant with gut dysbiosis, systemic inflammation, and intestinal barrier dysfunction [127]. These findings strongly support the hypothesis of a direct interaction between chronic microbial infections and central amyloidogenic processes in Alzheimer’s disease.

### 4.5. Oral–Gut Interactions in Alzheimer’s Disease

Preclinical models consistently show that periodontitis or oral pathogen exposure perturbs the intestinal ecosystem and barrier, producing gut dysbiosis, systemic inflammation, and downstream brain effects. For example, ligature-induced chronic periodontitis and repeated oral exposure to *P. gingivalis* produce gut microbiota shifts, intestinal barrier impairment, increased circulating LPSs and pro-inflammatory cytokines, and activation of innate pathways (i.e., NLRP3), culminating in BBB disruption, microglial activation, increased Aβ/tau pathology, and cognitive deficits in mice [128,129]. Clinical and population evidence complements the animal data: severe periodontitis has been associated with altered fecal microbiota profiles and depletion of short-chain fatty-acid producers in patients, and epidemiological cohorts report an increased risk of cognitive decline and Alzheimer’s disease among persons with long-standing periodontal disease [59,130,131]. Moreover, observational studies link periodontal inflammation with higher systemic inflammatory burden, a plausible mediator between oral dysbiosis, gut barrier dysfunction, and neuroinflammation in humans [132]. Importantly, recent work suggests that oral–gut communication is bidirectional. Gut dysbiosis itself can promote retrograde microbial influences on the oral niche, enabling gut-to-oral microbial seeding and altering salivary microbial composition. Experimental and clinical studies show that intestinal dysbiosis impairs mucosal immunity and increases luminal translocation of bacteria and metabolites, which can reach the oral cavity via hematogenous routes or enteral circulation, thereby reshaping oral microbial communities and reinforcing periodontal inflammation [4,22,52,53]. This feedback loop implies that oral dysbiosis not only drives gut dysfunction but may also be perpetuated by it, creating a self-amplifying cycle of microbial imbalance and systemic inflammation. Taken together, these preclinical and clinical lines of evidence support a model in which periodontitis-driven oral dysbiosis perturbs the gut microbiota and barrier, amplifies systemic inflammation and innate immune activation, and thereby promotes neuroinflammatory and amyloidogenic processes that accelerate cognitive decline. These findings align with broader evidence of gut-derived microbial metabolites and immune mediators modulating central neuroinflammation, supporting the integrated microbiome–gut–brain paradigm [133].

### 4.6. Oral Microbiota–Genetic Interactions in Alzheimer’s Disease

Emerging evidence indicates that oral microbiota dysbiosis, particularly the presence of periodontal pathogens such as *Treponema denticola* and *Tannerella forsythia*, may interact with genetic risk factors, including the *ApoE4* allele, to influence the development and progression of AD. These pathogens have been detected in the oral cavities of AD patients [134], where their presence correlates with lower cognitive scores and alterations in systemic immune biomarkers, such as neopterin and kynurenine. The *ApoE4* allele is a well-established genetic risk factor for AD, modulating cerebral lipid metabolism and amyloid-β clearance [135,136]. Evidence suggests that carriers of *ApoE4* who harbor oral pathogens may experience exacerbated systemic inflammation and neurodegenerative processes, indicating a synergistic effect between genetic susceptibility and periodontal infection [85]. Furthermore, single-nucleotide polymorphisms (SNPs) in genes such as VDR (Vitamin D Receptor) have been associated with increased susceptibility to both periodontitis and AD, highlighting a potential genetic link between oral health and neurodegeneration [137,138].

In addition to identifying microbial species associated with AD, Mendelian randomization (MR) analyses have leveraged SNPs to explore the genetic basis linking oral microbiota to AD [26,117]. Seven species, including *Centipeda periodontii*, *Haemophilus parainfluenzae*, *Campylobacter rectus*, *Leptotrichia massiliensis*, *Prevotella conceptionensis*, *Lachnoanaerobaculum*, and *Alloprevotella tannerae*, were significantly positively associated with AD (OR > 1, *p* ≤ 0.05), suggesting a promotive role in AD pathophysiology. Broad microbial associations were also observed for the *Saccharimonadaceae* family, *Streptococcus genus*, and *Solobacterium genus* indicating that diverse oral microbes may contribute to AD development [27]. Specifically, *P. gingivalis* demonstrated a strong positive association with AD (OR = 1.384, 95% CI: 1.021–1.878, *p* = 0.036), corroborating its proposed causative role. *Prevotella* (OR = 1.288, 95% CI: 1.082–1.534, *p* = 0.005) and *Treponema* (OR = 1.188, 95% CI: 1.028–1.374, *p* = 0.020) were similarly associated with increased AD risk [27], aligning with previous evidence linking these pathogens to periodontitis severity [139,140,141,142]. Interestingly, oral *Campylobacter* showed a protective effect (OR = 0.757, 95% CI: 0.612–0.937, *p* = 0.011), whereas *Campylobacter rectus* increased AD risk (OR = 1.277, 95% CI: 1.035–1.455, *p* = 0.018), highlighting species-specific effects [27]. The genus *Veillonella* was associated with decreased AD risk (OR = 0.725, 95% CI: 0.616–0.854, *p* < 0.001) [27]. Additionally, these MR findings are consistent with previous reports implicating genera such as *Eubacterium*, *Bacteroides*, *Porphyromonas*, *Fusobacterium*, and *Neisseria* in AD risk [143], with some overlap observed with alcohol-related dysbiosis. Notably, *Fusobacterium* showed a positive association with AD in the MR study (OR = 1.350, 95% CI: 1.036–1.759, *p* = 0.026) [27], supporting in vitro evidence linking this genus to amyloid adhesion and AD-like pathology [144]. Overall, the diverse molecular, microbial, and immunological pathways described above highlight the inherently multifactorial architecture of Alzheimer’s disease. Table 1 integrates these findings by outlining the major pathogenic factors and mechanistic domains currently implicated in AD development.

### 4.7. Epidemiological and Clinical Evidence

From an epidemiological standpoint, the accumulated evidence of recent years is consistent. Large-scale analyses, such as that by Kulkarni et al. [17], demonstrate that poor oral health doubles the risk of developing Alzheimer’s disease, while tooth loss is associated with more than a threefold risk increase. Further studies reinforce the concept that not only tooth extraction but also pulp vitality loss significantly increases the risk of Alzheimer’s disease, with particularly pronounced effects in the younger elderly population [18]. Observational clinical studies have also reported that patients with moderate-to-severe periodontitis and mild cognitive impairment are more likely to exhibit cerebral Aβ accumulation, as measured by PET imaging [145]. This suggests that periodontal disease may act as an early factor in the pathogenic cascade. Animal studies provide additional consistent evidence: dental extraction in Alzheimer’s transgenic mice accelerates neurodegeneration along the trigeminal–locus coeruleus–hippocampus pathway, leading to rapid memory decline [19,20,146]. Together, these observations not only reinforce the epidemiological association but also provide biological and temporal plausibility.

### 4.8. Oral Biomarkers and Salivary Diagnostics

A particularly promising field concerns the identification of easily accessible biomarkers, and saliva is emerging as a non-invasive diagnostic fluid. Multi-omics studies have revealed significant alterations in the microbial and metabolic composition of Alzheimer’s patients, including increased levels of *P. gingivalis* and *Veillonella parvula*, as well as metabolic shifts in molecules such as L-tyrosine and galactinol [24]. Another salivary biomarker of interest is lactotransferrin, an antimicrobial protein that is reduced in Alzheimer’s patients, reflecting diminished defense against oral and systemic infections [26]. Gingival crevicular fluid (GCF), a serum-derived exudate from the gingival sulcus, has also shown diagnostic potential, as it allows sensitive detection of local inflammatory mediators and periodontal immune activity in a minimally invasive manner [24]. Recent studies have further expanded the list of salivary biomarkers associated with AD. Elevated levels of salivary phospho-tau 181 (pTau181) have been observed in edentulous patients, suggesting a potential link between tooth loss and the progression of dementia [147]. pTau181 levels were significantly higher in the saliva and plasma of edentulous patients compared to controls without advanced periodontal disease, indicating that salivary pTau181 could serve as a novel associative marker for dementia [147]. Furthermore, salivary interleukin-34 (IL-34) levels are elevated in AD patients and are inversely associated with Mini-Mental State Examination (MMSE) scores, indicating a potential role in cognitive decline [148]. IL-34 is a cytokine involved in inflammation, and its increased presence in saliva may reflect systemic inflammatory processes linked to AD pathology. Despite these promising findings, standardized protocols for saliva collection and analysis are still lacking, representing a major barrier to clinical application [24]. Establishing consistent methodologies will be crucial for the widespread adoption of salivary biomarkers in AD diagnostics.

### 4.9. Therapeutic Perspectives

New therapeutic perspectives are emerging from the recognition of oral health as a determinant of Alzheimer’s disease. Several pilot clinical studies have shown that intensive oral hygiene programs can modestly alter the oral microbiota composition in elderly patients with dementia, suggesting that personalized interventions may enhance efficacy [25]. A recent 24-week randomized controlled trial evaluated the effects of 0.2% chlorhexidine gluconate in 100 patients with mild AD (66 completed) to determine effects oral microbiota dysbiosis and AD-related risk factors [149]. At baseline, poorer oral health correlated with fewer teeth, lower cognitive scores (MMSE), and higher diabetes prevalence. Chlorhexidine swabbing reduced AD- and periodontal-associated genera (*Porphyromonas*, *Treponema*, *Tannerella*) and improved microbial diversity, though cognitive scores were unchanged [149]. These findings suggest antiseptic interventions may restore oral microbial balance and reduce systemic inflammatory risk, highlighting *P. gingivalis* and *Treponema* as targets for future studies.

On the pharmacological and experimental front, various strategies are under investigation. Antimicrobial peptides such as β-defensin 3 have been shown to reduce Pg-LPS-induced IL-1β production in microglia, thereby attenuating the inflammatory response [150]. Phytochemical interventions have also shown efficacy in preclinical periodontal models: *Juglans regia* and *Pfaffia paniculata* extracts exhibited antimicrobial, anti-inflammatory, and immunomodulatory effects against *P. gingivalis* and related pathogens, potentially lowering Alzheimer’s risk by reducing peripheral inflammatory burden [151]. Caspase-4 inhibition, a key step in non-canonical inflammasome activation, has been shown to have neuroprotective effects by reducing oxidative stress and Aβ production [84]. Another approach targets gingipains, the cysteine proteases produced by *P. gingivalis*. A systematic review of preclinical studies highlights that gingipain inhibitors can reduce periodontal inflammation, bacterial load, and associated systemic effects, making them a promising adjunct in modulating host–microbe interactions relevant to neurodegeneration [152]. Likewise, mesenchymal stem cells of dental origin are emerging as innovative therapeutic resources, with both neuroprotective and regenerative potential in animal models [153]. Finally, a nanoparticulate vaccine against *P. gingivalis* has been proposed, aiming to prevent chronic infection and thereby reduce one of the major risk factors for Alzheimer’s disease [154].

In parallel with oral-targeted interventions, growing attention has shifted toward therapeutic strategies acting on the gut microbiota along the microbiome–gut–brain axis. Randomized controlled trials and meta-analyses show that probiotic supplementation can improve systemic inflammation, metabolic status, and, in some studies, cognitive performance in individuals with mild cognitive impairment or early Alzheimer’s disease, supporting the concept of microbiome-based neuromodulation [155,156,157]. Likewise, adherence to Mediterranean- or MIND-style dietary patterns reshapes gut microbial communities, increases short-chain fatty acid production, and is consistently associated with reduced neuroinflammation and lower risk of cognitive decline [158,159]. At the more intensive end of the therapeutic spectrum, fecal microbiota transplantation (FMT) from healthy donors has been shown to improve cognitive outcomes and reduce AD-related pathology in preclinical models, and early clinical case series report similar benefits in cognitively impaired patients, although mechanistic causality remains to be clarified [160,161,162]. Together, these findings underscore the potential of coordinated oral–gut microbiome interventions to modulate systemic inflammation, restore barrier integrity, and attenuate neuroinflammatory signaling.

Artificial intelligence (AI) is also emerging as a powerful tool in periodontal-systemic health, enabling earlier diagnosis, risk prediction, and treatment personalization. By integrating multi-omics data, clinical parameters, and imaging, AI can detect early dysbiosis and forecast systemic disease progression, including Alzheimer’s and cardiovascular conditions, improving diagnostic speed (~40%) and accuracy (~25%) [163]. AI-driven risk stratification allows tailored interventions, while longitudinal monitoring of microbiome and inflammatory markers can guide real-time treatment adjustments, including precision periodontal therapy, virulence-inhibiting agents, or anti-inflammatory regimens.

In a broader perspective, integrative therapies, such as improved oral hygiene regimens, probiotics or prebiotics to rebalance the microbial community, and host-directed anti-inflammatory agents, may act synergistically to reduce systemic inflammation and neuroinflammatory stimuli.

### 4.10. Future Directions

Future research should prioritize longitudinal human cohorts integrating periodontal status, gut/oral microbiome profiling, and cognitive trajectories to clarify temporal and causal pathways in the oral–gut–brain axis [4,22,52,53,59,127,128,129,130]. Equally crucial is the standardization of salivary, GCF, and fecal biomarker protocols, as current heterogeneity in sampling, sequencing platforms, and analytic pipelines limits reproducibility and clinical translation. Advances in personalized microbial therapeutics, including precision probiotics, targeted dietary modulation, and individualized microbiome-based interventions, hold promise for tailoring treatment to host genetics, immune signatures, and microbial profiles. Integrating multi-omics datasets with machine-learning approaches will be essential to identify reliable biomarkers, stratify risk, and optimize interventions aimed at modifying AD progression through oral–gut–brain axis modulation.

## 5. Discussion

The evidence synthesized in this review highlights the growing recognition of oral dysbiosis and periodontal disease as contributors to the onset and progression of Alzheimer’s disease. While the classical amyloid hypothesis has long dominated the field, mounting data suggest that chronic peripheral inflammation [76,77,79,81,82,83], microbial virulence factors [54,55,56,57,58,60], and systemic immune activation [77,78,79,80,81,82] play equally significant roles in neurodegeneration. This perspective reframes the oral cavity not as an isolated site of pathology but as a key regulator within the broader oral–gut–brain axis [22,126,127,128,130,131].

A recurring theme across molecular, experimental, and epidemiological studies is the central role of keystone pathogens such as *Porphyromonas gingivalis* and *Fusobacterium nucleatum*. Their ability to disrupt host immune responses [76,77,78,79,80,81], penetrate the blood–brain barrier [107,109], and modulate amyloid and tau pathology [113,114,115,116,117,118,119,120,121,122] supports a causal relationship rather than a mere association. However, the heterogeneity of oral microbiota communities and their dynamic interactions complicate efforts to define a single “Alzheimer-associated microbial signature”. Indeed, commensals may act as accessory pathogens under certain ecological conditions, underscoring the need for longitudinal and systems-level analyses [42,43].

Mechanistically, multiple overlapping pathways appear to mediate the link between oral dysbiosis and AD. These include chronic neuroinflammation [79], oxidative stress [94], ferroptosis [103], mitochondrial dysfunction [88,89,90], BBB disruption [105,106], and amyloid cross-seeding [125]. Rather than acting independently, these processes likely converge to amplify neurodegeneration in genetically susceptible individuals, particularly *ApoE4* carriers [135,136]. This multifactorial interplay suggests that future research should adopt an integrative approach, combining microbiome profiling with host genetic, immunological, and metabolic factors to better define risk stratification models.

From a translational perspective, salivary and gingival crevicular fluid biomarkers offer promising non-invasive diagnostic tools [24,147,148]. Early studies demonstrate alterations in antimicrobial proteins (e.g., lactotransferrin), cytokines, and microbial composition in AD patients. Yet, the lack of standardization in sample collection and analytic methods remains a major barrier to clinical implementation. Rigorous validation in large, multi-ethnic cohorts will be essential to establish reproducibility and clinical utility.

The therapeutic implications of these findings are particularly relevant. Pilot trials on antimicrobial mouth rinses, oral hygiene interventions [25,147], and natural compounds [151] suggest that targeted manipulation of the oral microbiome may mitigate systemic inflammatory burden, although cognitive benefits remain unproven and require larger randomized trials [150]. Emerging approaches, including gingipain inhibitors [152], caspase-4 blockade [84], mesenchymal stem cell therapies [153], and even nanoparticulate vaccines against *P. gingivalis* [154], offer promising avenues for modifying upstream inflammatory signals. In parallel, increasing attention has shifted toward gut-directed therapies along the microbiome–gut–brain axis. Probiotic supplementation has shown modest improvements in cognition and inflammatory/metabolic profiles in patients with mild cognitive impairment or AD [156,157,164]. Likewise, adherence to Mediterranean- or MIND-like diets, rich in fiber and polyphenols, beneficially shapes gut microbial communities and is consistently associated with reduced dementia risk [158,159,165,166]. More intensive interventions such as fecal microbiota transplantation improve cognition and reduce AD-like pathology in preclinical models, with early clinical reports suggesting similar trends in cognitively impaired patients, though causality remains to be demonstrated [160,161,162]. Overall, these oral- and gut-targeted strategies may be most effective when applied early, before extensive neurodegeneration occurs. Taken together, these therapeutic avenues support integrated approaches in which oral hygiene optimization, dietary modulation, and microbiome-directed interventions act synergistically to reduce systemic inflammation, stabilize the oral–gut–brain axis, and attenuate neuroinflammatory processes relevant to Alzheimer’s progression.

Despite compelling associations, several limitations temper current conclusions. Importantly, no direct causal relationship between periodontal disease or oral dysbiosis and Alzheimer’s disease has been demonstrated in humans. The association remains strongly suggestive but is influenced by multiple confounders, including diet, systemic health, socioeconomic status, medication use, and aging-related immune changes, which require careful control in future research. Most clinical studies remain cross-sectional or observational, limiting causal inference. Animal models provide mechanistic insights but do not fully replicate the complexity of human oral ecology and neurodegeneration. Additionally, publication bias favoring positive associations cannot be excluded.

In light of these gaps, future research priorities should include: (i) longitudinal human studies to establish temporal relationships between oral dysbiosis and cognitive decline; (ii) standardized, multi-omics approaches to identify robust microbial and salivary biomarkers; (iii) interventional trials assessing whether oral health improvements can alter AD trajectories; and (iv) integration of artificial intelligence and machine learning to analyze complex datasets and generate predictive models.

Taken together, the mouth–brain connection in AD represents a paradigm shift with profound clinical and public health implications. By positioning oral health as a modifiable risk factor, these findings not only expand our understanding of AD pathophysiology but also highlight opportunities for prevention and early intervention in an otherwise devastating and currently incurable disease.

## 6. Conclusions

The available evidence increasingly demonstrates that oral health, particularly periodontitis and oral microbiota dysbiosis, should not be regarded as mere comorbidities, but rather as risk factors that actively contribute to the pathogenesis of Alzheimer’s disease. The mechanisms involved, ranging from microglial activation and neuroinflammation to oxidative stress and ferroptosis, and extending to blood–brain barrier disruption and amyloidogenic processes, outline a complex yet coherent framework. Epidemiological studies consistently confirm that tooth loss and periodontal disease significantly increase the risk of Alzheimer’s disease. At the same time, emerging research highlights saliva and gingival crevicular fluid as promising sources of biomarkers for early diagnosis. These insights guide research toward innovative preventive and therapeutic strategies, ranging from the promotion of oral hygiene to the development of pharmacological and vaccine-based approaches targeting oral pathogens. Recognizing oral health as a modifiable risk factor, therefore, represents a tangible opportunity to mitigate the impact of Alzheimer’s disease and foster a more integrated, multidisciplinary clinical approach. Progress in this field will require interdisciplinary integration of dental medicine, neurology, immunology, nutrition science, and microbiome research to develop holistic preventive and therapeutic strategies for aging populations.

## Figures and Tables

**Figure 1 microorganisms-13-02741-f001:**
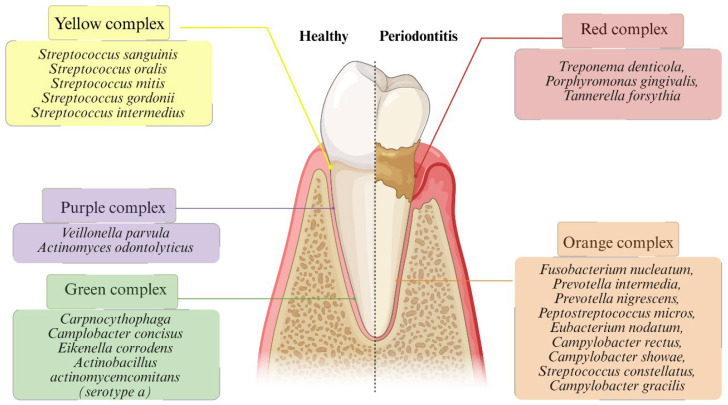
The Periodontal Microbial Complex Theory. Interactions among key bacterial species contribute to dysbiosis, periodontal tissue destruction, and systemic inflammation. Adapted from Socransky et al. [49].

**Table 1 microorganisms-13-02741-t001:** Summary of major mechanistic pathways linking oral dysbiosis and periodontal pathogens to AD. The table integrates key factors, principal findings, underlying mechanisms, and types of evidence, highlighting how microbial, inflammatory, and barrier-disrupting processes interact with classical AD pathology. Clinical implications reflect potential diagnostic and therapeutic applications, emphasizing oral health as a modifiable risk factor within the broader oral–gut–brain axis.

Factor	Main Findings	Mechanism	Type of Evidence	Clinical Implications
Amyloid hypothesis	Classical model of AD pathology	β-amyloid accumulation, extracellular plaques, tau hyperphosphorylation	Molecular/Neuropathological	Basis for drug development (anti-amyloid, anti-tau therapies)
Oral dysbiosis/Periodontitis	Chronic infection increases AD risk	Bacterial translocation, systemic endotoxins, immune dysregulation	Epidemiological + Molecular	Oral health as a modifiable risk factor
Peripheral amyloidogenesis in oral biofilm	Oral biofilms as sites of amyloid formation	Bacterial amyloids mimic or seed cerebral Aβ aggregation	Molecular + Microbiological	Suggests role of oral biofilm in systemic amyloid burden
*Porphyromonas gingivalis*	Keystone pathogen linked to AD	Pg-LPSs and gingipains activate microglia and astrocytes → neuroinflammation, oxidative stress, synaptic loss	Molecular + Experimental	Potential therapeutic target (inhibitors, vaccines)
*Fusobacterium nucleatum*	Contributes to AD pathology in models	Promotes Aβ accumulation and tau phosphorylation	Experimental (animal studies)	Supports bacterial link with neurodegeneration
Oxidative stress and Ferroptosis	Promote neuronal damage in AD	ROS imbalance, lipid peroxidation, ferroptotic pathways	Molecular	Antioxidant and anti-ferroptosis strategies
Blood–brain barrier impairment	Facilitates entry of bacterial products & toxins	Altered permeability, inflammation-driven endothelial dysfunction	Molecular + Experimental	Target for neuroprotective therapies
Oral–gut–brain axis	Oral microbiota interacts with intestinal microbiome	Metabolites, extracellular vesicles, inflammatory mediators alter BBB permeability and glial activation	Molecular + Experimental	Rationale for microbiome-targeted interventions

## Data Availability

No new data were created or analyzed in this study. Data sharing is not applicable to this article.

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
