# Peer review of "The Oral–Brain Axis in Alzheimer’s Disease: From Microbial Dysbiosis to Neurodegeneration"

_microorganisms, 2025, doi:10.3390/microorganisms13122741_

Round 1

Reviewer 1 Report

Comments and Suggestions for Authors

This review provides a well-structured, detailed, and mechanistic overview of how oral dysbiosis contributes to Alzheimer’s disease (AD) pathophysiology. The manuscript is scientifically sound, highly referenced, and logically organized.

Abstract

The abstract is dense and lists mechanisms without emphasizing the conceptual novelty or the gap this review fills. A clearer concluding statement defining how this review advances current understanding would strengthen its impact.

Introduction

  • The transition from classical AD hypotheses to microbial dysbiosis could be smoother—currently abrupt. Suggest adding a bridging paragraph connecting systemic inflammation to oral microbiota and brain outcomes.
  • The introduction focuses primarily on local oral inflammation; it would benefit from situating oral dysbiosis within the broader microbiome–gut–brain concept.

  1. Oral Microbiome and Periodontal Complex
  • Add a short paragraph highlighting how microbial community shifts influence systemic inflammation beyond oral tissues—linking to the bidirectional nature of microbiome–brain signaling.
  1. Oral Dysbiosis and Neurodegeneration
  • Section 4.5 (Oral–gut interactions) could be expanded to include bidirectional feedback (gut-to-oral microbial seeding).
  • After line 378 (“…promotes neuroinflammatory and amyloidogenic processes that accelerate cognitive decline”): “These findings align with broader evidence of gut-derived microbial metabolites and immune mediators modulating central neuroinflammation, supporting the integrated microbiome–gut–brain paradigm. https://doi.org/10.1007 /s12035-025-04846-0

Discussion

  • Expand on therapeutic strategies that target both oral and gut microbiota, such as probiotics, diet modification, or microbiome transplants.
  • Add a concise Future Directions paragraph—focusing on longitudinal studies, standardization of biomarkers, and personalized microbial therapeutics.

Author Response

We sincerely thank the Reviewer for the careful evaluation and the constructive comments, which have significantly improved the clarity and scientific quality of the manuscript. We have addressed all suggestions as follows:

  • Abstract: We revised the final section to clearly highlight the conceptual novelty of the review and the specific knowledge gap it addresses.

  • Introduction: We added a bridging paragraph to ensure a smoother transition from classical AD hypotheses to microbial dysbiosis and expanded the introduction to better situate oral dysbiosis within the broader microbiome-gut-brain axis.

  • Oral Microbiome and Periodontal Complex: We incorporated a concise paragraph explaining how microbial community shifts contribute to systemic inflammation and bidirectional microbiome-brain signaling.

  • Oral Dysbiosis and Neurodegeneration: Section 4.5 was expanded to include bidirectional oral-gut interactions. We also inserted the requested statement regarding gut-derived metabolites and their role in neuroinflammation, including the recommended citation.

  • Future Directions paragraph was added, highlighting the need for longitudinal studies, biomarker standardization, and personalized microbial therapeutics.

  • Discussion: We expanded the therapeutic section to include strategies targeting both oral and gut microbiota (probiotics, dietary modulation, and fecal microbiota transplantation).

Reviewer 2 Report

Comments and Suggestions for Authors
  • Recent research indicates that periodontal diseases may contribute to the pathogenesis of Alzheimer's disease (AD) through various mechanisms, including microbial invasion, systemic inflammation, oral microbiota-genetic and gut interactions.
  • However, the relationship between oral microbiota and AD or cognitive decline is complex and multifaceted. Although periodontal pathogenic bacteria have been detected in brain tissue and cerebrospinal fluid of patients with AD, a direct causal link has not yet been demonstrated in human clinical studies.
  • Periodontal diseases may be a modifiable risk factor for AD, and periodontal interventions could contribute to AD prevention and management. Further clinical studies are needed to confirm the diagnostic biomarkers and methods, as well as the therapeutic potential of targeting oral microbiota in AD.
  • Future research should prioritize systemic studies to establish causal relationships and elucidate the temporal dynamics between periodontal disease and Alzheimer’s disease. The importance of interdisciplinary approaches that integrate dental, neurological, and nutritional care should also be emphasized in the review to develop holistic solutions to treat AD in aging populations.

Author Response

We thank the Reviewer for these valuable observations. We have revised the manuscript to emphasize the multifactorial and non-causal nature of the relationship between periodontal disease, oral microbiota, and Alzheimer’s disease. We also highlighted periodontal disease as a potential modifiable risk factor and clarified the need for additional clinical studies to validate biomarkers and therapeutic approaches targeting the oral microbiota.

A concise Future Directions section has been added, stressing the importance of longitudinal studies to define temporal and causal relationships, as well as the relevance of interdisciplinary approaches integrating dental, neurological, and nutritional perspectives. All suggestions have been fully incorporated into the revised manuscript.